

# Association between the HTR2C rs1414334 C/G gene polymorphism and the development of the metabolic syndrome in patients treated with atypical antipsychotics

José María Rico-Gomis[1,2], Antonio Palazón-Bru[1,3], Irene Triano-García[4], Luis Fabián Mahecha-García[2], Ana García-Monsalve[4], Andrés Navarro-Ruiz[4], Berta Villagordo-Peñalver[2], Jessica Jiménez-Abril[2], Alicia Martínez-Hortelano[2] and Vicente Francisco Gil-Guillén[1,3]

[1] Department of Clinical Medicine, Miguel Hernández University, San Juan de Alicante, Alicante, Spain
[2] Department of Psychiatry, General University Hospital of Elche, Elche, Alicante, Spain
[3] Research Unit, General University Hospital of Elda, Elda, Alicante, Spain
[4] Pharmacy Service, General University Hospital of Elche, Elche, Alicante, Spain

## ABSTRACT

Few studies have assessed the association between the rs1414334 C/G polymorphism in the HTR2C gene and the development of the metabolic syndrome in patients treated with atypical antipsychotics. To provide further evidence, a cross-sectional study was conducted in Spain between 2012 and 2013 in 166 patients with these characteristics. In these patients, the association between the polymorphism and the presence of the metabolic syndrome was determined by implementing binary logistic regression models adjusted for variables associated with the metabolic syndrome. We did not confirm previous claims that the C allele of the polymorphism was linked to the metabolic syndrome: the association was in the opposite direction and non-significant. This conclusion held after taking gender and lifestyle variables into account.

Corresponding author
Antonio Palazón-Bru,
antonio.pb23@gmail.com

## INTRODUCTION

Cardiovascular disease is the leading cause of mortality in the world (*World Health Organization*, *2014*). Its main risk factors are hypertension, diabetes mellitus and dyslipidemia, which are grouped together in the definition of the metabolic syndrome (*Alberti et al.*, *2005*; *Grundy et al.*, *2004*). This syndrome occurs more frequently in schizophrenic patients (*McEvoy et al.*, *2005*; *Meyer & Stahl*, *2009*), leading to an increase in cardiovascular risk (*Osby et al.*, *2000*; *Casey et al.*, *2004*).

In this type of patient, taking atypical antipsychotics can cause an increase in the probability of having the metabolic syndrome (*Newcomer*, *2004*; *McEvoy et al.*, *2005*). The weight gain associated with the metabolic syndrome is related to both environmental and

genetic factors (*Gebhardt et al., 2010*). Therefore, identifying relevant genetic variants would be useful when choosing personalized treatments, minimizing the risk of cardiovascular disease in those patients genetically predisposed to developing the metabolic syndrome.

Among the genes involved, previous studies have reported an association between specific polymorphisms in the HTR2C gene and development of the metabolic syndrome in these patients (*Mulder et al., 2007*; *Mulder et al., 2009*; *Risselada et al., 2012*; *Ma et al., 2014*). 5-HT2C agonists decrease appetite (*Clifton, Lee & Dourish, 2000*), while antagonists increase appetite (*Bonhaus et al., 1997*). In addition, it appears that 5-HT2C receptors are related to the anorexigenic effects of leptin (*Von Meyenburg, Langhans & Hrupka, 2003*; *Reynolds, Hill & Kirk, 2006*).

For this reason, multiple genetic variants of the HTR2C gene have been studied. Various polymorphisms such as –759C, –697C/G, –997G/A, –1,165A/G have been evaluated, finding associations with the development of the metabolic syndrome. At times these associations have shown conflicting results (*Wallace et al., 2011*). Our focus was on the rs1414334 C/G polymorphism in the HTR2C gene, because in our literature search we found only 3 studies, conducted in the Netherlands, of which two were replication studies (*Mulder et al., 2007*; *Mulder et al., 2009*; *Risselada et al., 2012*). In the first study (*Mulder et al., 2007*), among patients receiving treatment with atypical antipsychotics, 10 out of 21 (48%) with the C allele had the metabolic syndrome compared with 18 out of 91 (20%) with the G allele. In the first replication study (*Mulder et al., 2009*), 13 in 31 patients with the C allele had the metabolic syndrome (42%), and 39 in 133 patients with the G allele had the syndrome (29%). Finally, in the second replication study (*Risselada et al., 2012*), 16 out of 30 patients with the C allele had the metabolic syndrome (53%), compared with 40 out of 132 (30%) with the G allele. Later, a meta-analysis included these three studies and determined a significant association between the presence of the C allele of the rs1414334 polymorphism and the prevalence of the metabolic syndrome in patients receiving treatment with atypical antipsychotics (*Ma et al., 2014*). On the other hand, HTR2C is X-linked, and so females can be heterozygous whereas males have only one copy of the gene. Data from these previous studies did not, however, break down results by gender studies (*Mulder et al., 2007*; *Mulder et al., 2009*; *Risselada et al., 2012*).

This behavior has also been noted in the 759C/T and 997G/A polymorphisms in the HTR2C gene, where some authors have found these polymorphisms to have a protective nature against the metabolic syndrome, whilst others have found them to be of risk and the remainder found no difference (*Wallace et al., 2011*; *Ma et al., 2014*). Additionally, these authors found that other polymorphisms of the 5-HT2C gene (not rs1414334) were associated with weight gain in earlier stages of treatment (*Wallace et al., 2011*). In other words, in the scientific literature contradictory or null results in different polymorphisms of the HTR2C gene are reported (*Ma et al., 2014*). Since in the rs1414334 polymorphism the results were consistent, but were always analyzed by the same research group in the same region (*Mulder et al., 2007*; *Mulder et al., 2009*; *Risselada et al., 2012*), we decided to ascertain whether their results agreed with those obtained in other geographical areas, such as Spain. For this reason, to provide further evidence for the possible association between this polymorphism and the metabolic syndrome in psychotic patients, we conducted a

study in Spain. We aimed to replicate the previous finding of an association between the presence of the C allele of rs1414334 and the metabolic syndrome in patients treated with atypical antipsychotics. We extended previous work by considering gender effects on the association, and by incorporating measures of relevant lifestyle variables, such as diet.

## MATERIALS & METHODS

### Setting

The proposed study was conducted in the Elche, Crevillente and Santa Pola Department of Health of the Valencian Community, which is a Mediterranean region located in southeastern Spain. In its mental health services, coverage is provided to approximately 325,000 inhabitants (almost entirely Caucasian). Health care is universal and free for all patients, except those with an irregular immigration status.

### Study design and participants

In this cross-sectional, observational study, a sample of inpatients and outpatients treated for at least three months with atypical antipsychotics (clozapine, olanzapine, risperidone, quetiapine, aripiprazole, ziprasidone, amisulpride, asenapine and paliperidone) was analyzed. For the sample, consecutive patients attending Mental Health Services of the Valencian Community at the Elche, Crevillente and Santa Pola Department of Health between December 2012 and June 2013 were selected. All patients had to be of legal age (>18 years) and have a diagnosis of a psychotic disorder. This was defined as having a psychiatrist's diagnosis of schizophrenia, schizophreniform disorder, schizoaffective disorder or other psychotic disorders, or bipolar disorder, and continuous treatment with antipsychotic medication (*American Psychiatric Association*, *2000*). To comply with the current legislation, all patients were adequately informed about the study (written form) and, if agreeing to participate, signed a consent form before inclusion.

The study was approved by the Clinical Research Ethics Committee of the General University Hospital of Elche on November 20, 2012.

### Variables and measurements

The primary variable was the diagnosis of the metabolic syndrome using the ATPIII criteria (*Grundy et al.*, *2004*), i.e., the presence of three or more of the following metabolic criteria: abdominal obesity, hypertriglyceridemia, low levels of high density lipoprotein cholesterol (HDL-C), elevated blood pressure and insulin resistance. All parameters of the metabolic syndrome criteria were determined using appropriate clinical guidelines (*NHLBI Obesity Education Initiative Task Force*, *1998*; *NCEP*, *2002*; *Mancia et al.*, *2007*; *American Diabetes Association*, *2012*).

Secondary clinical variables were: age (years), gender, diagnosis (schizophrenia or other) (*American Psychiatric Association*, *2000*), years with the diagnosis, treatment duration with the current antipsychotic (years), continuing therapy with a favorable (aripiprazole, ziprasidone or palperidone) or unfavorable (olanzapine, clozapine, risperidone, quetiapine or asenapine) (*American Diabetes Association et al.*, *2004*; *Stahl, Mignon & Meyer*, *2009*; *Schreiner et al.*, *2012*; *Stoner & Pace*, *2012*; *Fleischhacker et al.*, *2013*; *Young, Taylor &*

*Lawrie*, *2015*) metabolic profile, use of antidepressants, mood stabilizers and number of pack-years (number of cigarettes smoked per day × number of years smoked)/20). These variables were obtained by clinical interview with the patient and corroborated by the electronic medical record. It is obligatory that all drug prescriptions be made through this computerized system (*Agencia Valenciana de Salud*, *2011*).

The following variables related to lifestyle were obtained by conducting the National Survey of Health and the European Health Survey (*Ministerio de Sanidad, Política Social e Igualdad & Instituto Nacional de Estadística*, *2009*; *Ministerio de Sanidad, Servicios Sociales e Igualdad & Instituto Nacional de Estadística*, *2011*): usual number of hours of sleep, physical activity during leisure time and habitual activity (work and home) and consumption of food (fruit, meat, eggs, fish, pasta, rice and potatoes, bread and cereals, vegetables, legumes, cold meats, milk products and sweets). Regular physical activity was assessed on a scale as follows: 0, sitting; 1, standing; 2, walking or moving around; 3, great effort. Physical activity during leisure time was measured with the following scale: 0, never; 1, less than once per month; 2, once or several times per month; 3, one or more times per week. Finally, the scale of all the food (fruit, meat, eggs, fish, pasta, rice and potatoes, bread and cereals, vegetables, legumes, cold meats, milk products and sweets) was rated as: 0, daily; 1, three or more times per week; 2, once or twice per week; 3, less than once per week; 4, never or almost never.

## DNA isolation and genotyping

Genomic DNA was extracted from EDTA blood samples using the semi-automated QIAcube system (Qiagen, Hilden, Germany). The rs1414334 C/G polymorphism was analyzed by real-time PCR using a pre-developed assay (C_7455701_10, Applied Biosystems, Madrid, Spain) for allelic discrimination in a 7300 Real-Time PCR System (Applied Biosystems). The reaction was carried out with a TaqMan genotyping PCR master mix (Applied Biosystems), according to the protocol provided by Applied Biosystems.

The rs1414334 polymorphism allele C/G is described in the SNP database (http://www.ncbi.nlm.nih.gov/snp).

## Sample size

The sample size during the study period was 166 patients, 68 of whom had the metabolic syndrome. As the sample was collected a posteriori, the power of the test was calculated to establish an association between an exposure factor and a disease; that is, whether an odds ratio (OR) is different from 1. For this calculation we assumed C allele in 30% of cases and 13% of controls, a control/case ratio of 2:1 and a type I error of 5% (*Risselada et al.*, *2012*). With these parameters, the power of the test was 81.7%.

## Statistical methods

All variables were described using frequencies (absolute and relative), means and standard deviations. A stepwise logistic regression model was constructed to determine the relationship between the polymorphism and the metabolic syndrome. Since we had 68 events (metabolic syndrome), the model could only hold a maximum of six explanatory variables (one for every 10 events) (*Ramírez-Prado et al.*, *2015*). As we had a total of

**Table 1** Factors associated with metabolic syndrome in the patients diagnosed with psychiatric disorders in a Spanish region: total sample.

| Variable | Total $n = 166$ $n(\%)/x \pm s$ | With MS $n = 68(41.0\%)$ $n(\%)/x \pm s$ | p-value | Adj. OR (95% CI) | p-value |
|---|---|---|---|---|---|
| rs1414334 allele: | | | | | |
| C | 33(19.9) | 10(30.3) | 0.164 | 0.42(0.17–1.04) | 0.062 |
| G | 133(80.1) | 58(43.6) | | 1 | |
| Gender: | | | | | |
| Male | 99(59.6) | 45(45.5) | 0.153 | N/M | N/M |
| Female | 67(40.4) | 23(34.3) | | | |
| Age (years) | 43.1 ± 11.5 | 45.3 ± 11.0 | 0.038 | 1.03(0.99–1.06) | 0.158 |
| Schizophrenia: | | | | | |
| Yes | 92(55.4) | 41(44.6) | 0.293 | N/M | N/M |
| No | 74(44.6) | 27(36.5) | | | |
| Years with the disorder | 14.9 ± 9.6 | 16.4 ± 9.0 | 0.099 | 1.01(0.97–1.05) | 0.654 |
| Years with the current therapy | 5.3 ± 4.5 | 5.0 ± 4.7 | 0.594 | N/M | N/M |
| Therapy with a favorable metabolic profile: | | | | | |
| Yes | 49(29.5) | 19(38.8) | 0.711 | N/M | N/M |
| No | 117(70.5) | 49(41.9) | | | |
| Therapy with an unfavorable metabolic profile: | | | | | |
| Yes | 137(82.5) | 56(40.9) | 0.960 | N/M | N/M |
| No | 29(17.5) | 12(41.4) | | | |
| Antidepressants: | | | | | |
| Yes | 47(28.3) | 25(53.2) | 0.044 | 2.12(1.00–4.49) | 0.051 |
| No | 119(71.7) | 43(36.1) | | 1 | |
| Mood stabilizers: | | | | | |
| Yes | 37(22.3) | 17(45.9) | 0.485 | N/M | N/M |
| No | 37(22.3) | 51(39.5) | | | |
| Number of pack-years | 11.6 ± 17.5 | 14.8 ± 22.3 | 0.073 | 1.02(1.00–1.04) | 0.055 |
| Sleeping hours | 8.9 ± 1.9 | 8.9 ± 1.9 | 0.841 | N/M | N/M |
| Physical activity at work and home | 0.9 ± 0.8 | 0.8 ± 0.8 | 0.116 | 0.80(0.52–1.21) | 0.286 |
| Physical activity in leisure time | 1.1 ± 1.2 | 1.0 ± 1.2 | 0.222 | N/M | N/M |
| Fruit | 1.1 ± 1.2 | 1.2 ± 1.3 | 0.469 | N/M | N/M |
| Meat | 1.1 ± 0.9 | 1.0 ± 0.8 | 0.203 | N/M | N/M |
| Eggs | 1.8 ± 0.8 | 1.7 ± 0.8 | 0.336 | N/M | N/M |
| Fish | 2.0 ± 0.9 | 2.0 ± 0.9 | 0.398 | N/M | N/M |
| Pasta, rice and potatoes | 1.1 ± 0.9 | 1.1 ± 0.9 | 0.714 | N/M | N/M |
| Bread and cereals | 0.5 ± 0.9 | 0.4 ± 0.9 | 0.294 | N/M | N/M |
| Vegetables | 1.1 ± 1.2 | 1.3 ± 1.2 | 0.137 | N/M | N/M |
| Legumes | 1.8 ± 0.9 | 1.8 ± 1.0 | 0.643 | N/M | N/M |
| Cold meats | 2.1 ± 1.3 | 2.0 ± 1.3 | 0.539 | N/M | N/M |

**Table 1** (*continued*)

| Variable | Total<br>$n = 166$<br>$n(\%)/x \pm s$ | With MS<br>$n = 68(41.0\%)$<br>$n(\%)/x \pm s$ | *p*-value | Adj. OR<br>(95% CI) | *p*-value |
|---|---|---|---|---|---|
| Milk products | $0.7 \pm 1.0$ | $0.6 \pm 1.0$ | 0.622 | N/M | N/M |
| Sweets | $2.1 \pm 1.4$ | $2.1 \pm 1.4$ | 0.913 | N/M | N/M |

**Notes.**

Abbreviations: Adj. OR, adjusted odds ratio; CI, confidence interval; MS, metabolic syndrome; N/M, not in the model; $n(\%)$, absolute frequency (relative frequency); $x \pm s$, mean ± standard deviation.

Goodness-of-fit of the multivariate model: $X^2 = 16.96$, $p = 0.009$. Regular physical activity was assessed on a scale as follows: 0, sitting; 1, standing; 2, walking or moving around; 3, great effort. Physical activity during leisure time was measured using the following scale: 0, never; 1, less than once per month; 2, once or several times per month; 3, one or more times per week. The scale of all the food was rated as: 0, daily; 1, three or more times a week; 2, once or twice a week; 3, less than once a week; 4, never or almost never.

25 variables, to determine which would be introduced in the model we first compared proportions and means between the metabolic syndrome (yes and no) by using the Chi-square test and the $t$-test, and then selected those variables with the smaller $p$-value. Thus the adjusted OR were obtained. The goodness-of-fit of the final model was assessed using the likelihood ratio test.

Since the HTR2C gene is X-linked, all the analyses were carried out for men and women separately. This is important because women encounter a very different situation than men. Women can be heterozygote, whereas men cannot, which creates a completely different biological scenario. This is also important with respect to previous articles published on this topic (*Mulder et al.*, *2007*; *Mulder et al.*, *2009*; *Risselada et al.*, *2012*).

All analyses were performed with a significance of 5% and for each relevant parameter its associated confidence interval (CI) was calculated. The statistical package used was IBM SPSS Statistics 19 (IBM, Armonk, NY, USA).

## RESULTS

In the patients with the rs1414334 C allele, the prevalence of the metabolic syndrome was: total, 30.3%; men, 35.7%; and women, 26.3%. These values for the rs1414334 C allele were: total, 43.6%; men, 47.1%; and women, 37.5%. There were no differences between the polymorphism and our outcome ($0.164 < p$-value $< 0.430$) (Tables 1–3).

The descriptive and analytical characteristics of the total sample are shown in Table 1. We emphasize the prevalence of the C allele of the polymorphism of 19.9% and of the diagnosis of schizophrenia (55.4%), a high proportion of drugs with an unfavorable metabolic profile (82.5%) and an average disease duration close to 15 years. Regarding the differences in the diagnosis of the metabolic syndrome (Table 1), we found statistically significant differences in the following variables ($p < 0.05$): older age ($p = 0.038$) and antidepressants ($p = 0.044$). The rs1414334 C allele did not show any difference ($p = 0.164$). When we analyzed the factors associated with the metabolic syndrome (Table 1), no variables were significantly different, including the C allele of the polymorphism ($p = 0.062$) (direction contrary to prediction).

In the men (Table 2), we noted differences between an older age and the prevalence of the metabolic syndrome ($p = 0.011$). On the other hand, the logistic regression model only showed one significant factor associated with the metabolic syndrome (older age), and the

**Table 2  Factors associated with metabolic syndrome in the patients diagnosed with psychiatric disorders in a Spanish region: males.**

| Variable | Total $n=99$ $n(\%)/x \pm s$ | With MS $n=45(45.5\%)$ $n(\%)/x \pm s$ | p-value | Adj. OR (95% CI) | p-value |
|---|---|---|---|---|---|
| rs1414334 allele: | | | | | |
| C | 14(14.1) | 5(35.7) | 0.430 | 0.35(0.09–1.42) | 0.141 |
| G | 85(85.9) | 40(47.1) | | 1 | |
| Age (years) | 39.9 ± 8.9 | 42.4 ± 8.8 | 0.011 | 1.06(1.00–1.12) | 0.036 |
| Schizophrenia: | | | | | |
| Yes | 62(62.6) | 30(48.4) | 0.448 | N/M | N/M |
| No | 37(37.4) | 15(40.5) | | | |
| Years with the disorder | 14.5 ± 8.0 | 15.8 ± 7.9 | 0.166 | N/M | N/M |
| Years with the current therapy | 5.4 ± 4.6 | 5.2 ± 4.9 | 0.635 | N/M | N/M |
| Therapy with a favorable metabolic profile: | | | | | |
| Yes | 27(27.3) | 11(40.7) | 0.564 | N/M | N/M |
| No | 72(72.7) | 34(47.2) | | | |
| Therapy with an unfavorable metabolic profile: | | | | | |
| Yes | 83(83.8) | 39(47.0) | 0.485 | N/M | N/M |
| No | 16(16.2) | 6(37.5) | | | |
| Antidepressants: | | | | | |
| Yes | 22(22.2) | 13(59.1) | 0.145 | N/M | N/M |
| No | 77(77.8) | 32(41.6) | | | |
| Mood stabilizers: | | | | | |
| Yes | 25(25.3) | 15(60.0) | 0.091 | 2.16(0.79–5.91) | 0.134 |
| No | 74(74.7) | 30(40.5) | | 1 | |
| Number of pack-years | 13.9 ± 19.9 | 17.6 ± 25.3 | 0.119 | 1.01(0.99–1.04) | 0.418 |
| Sleeping hours | 8.9 ± 2.1 | 8.7 ± 2.1 | 0.432 | N/M | N/M |
| Physical activity at work and home | 0.9 ± 0.9 | 0.8 ± 0.8 | 0.140 | N/M | N/M |
| Physical activity in leisure time | 1.1 ± 1.2 | 1.0 ± 1.2 | 0.483 | N/M | N/M |
| Fruit | 1.3 ± 1.2 | 1.4 ± 1.3 | 0.231 | N/M | N/M |
| Meat | 1.0 ± 0.8 | 0.9 ± 0.7 | 0.488 | N/M | N/M |
| Eggs | 1.7 ± 0.9 | 1.5 ± 0.9 | 0.122 | N/M | N/M |
| Fish | 2.0 ± 0.9 | 2.1 ± 1.0 | 0.451 | N/M | N/M |
| Pasta, rice and potatoes | 1.1 ± 0.9 | 1.1 ± 0.9 | 0.685 | N/M | N/M |
| Bread and cereals | 0.4 ± 0.8 | 0.3 ± 0.8 | 0.287 | N/M | N/M |
| Vegetables | 1.4 ± 1.2 | 1.5 ± 1.2 | 0.374 | N/M | N/M |
| Legumes | 1.9 ± 0.9 | 1.9 ± 1.0 | 0.448 | N/M | N/M |
| Cold meats | 2.0 ± 1.2 | 1.9 ± 1.3 | 0.636 | N/M | N/M |
| Milk products | 0.7 ± 1.0 | 0.7 ± 1.0 | 0.805 | N/M | N/M |
| Sweets | 2.0 ± 1.4 | 1.9 ± 1.4 | 0.553 | N/M | N/M |

**Notes.**

Abbreviations: Adj. OR, adjusted odds ratio; CI, confidence interval; MS, metabolic syndrome; N/M, not in the model; $n(\%)$, absolute frequency (relative frequency); $x \pm s$, mean ± standard deviation.

Goodness-of-fit of the multivariate model: $X^2 = 10.96$, $p = 0.027$. Regular physical activity was assessed on a scale as follows: 0, sitting; 1, standing; 2, walking or moving around; 3, great effort. Physical activity during leisure time was measured using the following scale: 0, never; 1, less than once per month; 2, once or several times per month; 3, one or more times per week. The scale of all the food was rated as: 0, daily; 1, three or more times a week; 2, once or twice a week; 3, less than once a week; 4, never or almost never.

**Table 3  Factors associated with metabolic syndrome in the patients diagnosed with psychiatric disorders in a Spanish region: females.**

| Variable | Total $n = 67$ $n(\%)/x \pm s$ | With MS $n = 23(35.4\%)$ $n(\%)/x \pm s$ | p-value | Adj. OR (95% CI) | p-value |
|---|---|---|---|---|---|
| rs1414334 allele: | | | | | |
| C | 19(28.4) | 5(26.3) | 0.385 | 0.54(0.16–1.82) | 0.318 |
| G | 48(71.6) | 18(37.5) | | 1 | |
| Age (years) | 47.7 ± 13.3 | 51.0 ± 12.8 | 0.148 | N/M | N/M |
| Schizophrenia: | | | | | |
| Yes | 30(44.8) | 11(36.7) | 0.717 | N/M | N/M |
| No | 37(55.2) | 12(32.4) | | | |
| Years with the disorder | 15.4 ± 11.6 | 17.6 ± 10.9 | 0.269 | N/M | N/M |
| Years with the current therapy | 5.1 ± 4.3 | 4.8 ± 4.4 | 0.735 | N/M | N/M |
| Therapy with a favorable metabolic profile: | | | | | |
| Yes | 22(32.8) | 8(36.4) | 0.806 | N/M | N/M |
| No | 45(67.2) | 15(33.3) | | | |
| Therapy with an unfavorable metabolic profile: | | | | | |
| Yes | 54(80.6) | 17(31.5) | 0.344 | N/M | N/M |
| No | 13(19.4) | 6(46.2) | | | |
| Antidepressants: | | | | | |
| Yes | 25(37.3) | 12(48.0) | 0.069 | 2.74(0.95–7.90) | 0.063 |
| No | 42(62.7) | 11(26.2) | | 1 | |
| Mood stabilizers: | | | | | |
| Yes | 12(17.9) | 2(16.7) | 0.195 | N/M | N/M |
| No | 55(82.1) | 21(38.2) | | | |
| Number of pack-years | 8.2 ± 12.6 | 9.6 ± 14.3 | 0.517 | N/M | N/M |
| Sleeping hours | 8.9 ± 1.6 | 9.2 ± 1.5 | 0.348 | N/M | N/M |
| Physical activity at work and home | 1.0 ± 0.8 | 1.0 ± 0.8 | 0.648 | N/M | N/M |
| Physical activity in leisure time | 1.1 ± 1.1 | 0.9 ± 1.1 | 0.261 | N/M | N/M |
| Fruit | 0.9 ± 1.1 | 0.9 ± 1.2 | 0.732 | N/M | N/M |
| Meat | 1.2 ± 1.0 | 1.2 ± 0.9 | 0.958 | N/M | N/M |
| Eggs | 1.9 ± 0.8 | 2.0 ± 0.7 | 0.419 | N/M | N/M |
| Fish | 2.0 ± 0.8 | 2.0 ± 0.7 | 0.733 | N/M | N/M |
| Pasta, rice and potatoes | 1.2 ± 0.9 | 1.0 ± 0.9 | 0.363 | N/M | N/M |
| Bread and cereals | 0.7 ± 1.1 | 0.6 ± 1.1 | 0.834 | N/M | N/M |
| Vegetables | 0.7 ± 1.0 | 0.9 ± 1.1 | 0.454 | N/M | N/M |
| Legumes | 1.7 ± 0.9 | 1.6 ± 0.8 | 0.647 | N/M | N/M |
| Cold meats | 2.3 ± 1.4 | 2.3 ± 1.4 | 0.891 | N/M | N/M |
| Milk products | 0.7 ± 1.1 | 0.6 ± 1.1 | 0.617 | N/M | N/M |
| Sweets | 2.2 ± 1.5 | 2.4 ± 1.4 | 0.493 | N/M | N/M |

**Notes.**

Abbreviations: Adj. OR, adjusted odds ratio; CI, confidence interval; MS, metabolic syndrome; N/M, not in the model; $n(\%)$, absolute frequency (relative frequency); $x \pm s$, mean ± standard deviation.

Goodness-of-fit of the multivariate model: $X^2 = 4.31$, $p = 0.116$. Regular physical activity was assessed on a scale as follows: 0, sitting; 1, standing; 2, walking or moving around; 3, great effort. Physical activity during leisure time was measured using the following scale: 0, never; 1, less than once per month; 2, once or several times per month; 3, one or more times per week. The scale of all the food was rated as: 0, daily; 1, three or more times a week; 2, once or twice a week; 3, less than once a week; 4, never or almost never.

C allele of the polymorphism had the same association as was found in the total sample (inverse and non-significant, $p = 0.141$). Finally (Table 3), in the women there were no differences in the bivariate analysis (Chi-square test and the $t$-test), no significant factors associated with our outcome, and the studied polymorphism had the same result as in the total sample and in the men (inverse and non-significant, $p = 0.318$).

## DISCUSSION

### Summary

This study found no statistically significant association between the presence of the C allele of the rs1414334 polymorphism in the HTR2C gene and the metabolic syndrome.

### Study strengths and limitations

The main strength of this study is that it provides greater knowledge of the association between the polymorphism analyzed and the metabolic syndrome. Moreover, all the variables that the Dutch group did not analyze in their studies were taken into account, such as physical activity, diet and smoking (*Mulder et al.*, *2007*; *Mulder et al.*, *2009*; *Risselada et al.*, *2012*). To minimize information bias, all measurements were taken in direct contact with the patient and according to current clinical guidelines. To minimize selection bias, data were taken from all the patients who attended mental health services during a specific period (six months).

Regarding limitations, we must bear in mind that our design is cross-sectional, thus we could not establish causal relationships between the factors analyzed and the presence of the metabolic syndrome. In other words, our study, as in the Dutch group's study (*Mulder et al.*, *2007*; *Mulder et al.*, *2009*; *Risselada et al.*, *2012*; *Ma et al.*, *2014*), only assessed the association between the polymorphism studied and the metabolic syndrome; so to establish causality longitudinal studies should be performed. We must point out that the sample of our patients corresponds to a population treated with atypical antipsychotics, thus we cannot extrapolate the results to the general population (high heterogeneity). However, the aim of this study was to analyze the association between the metabolic syndrome and the polymorphism studied in patients with this type of treatment. Consequently, the results were obtained from a homogeneous sample of the population treated with atypical antipsychotics. Moreover, the development of the metabolic syndrome may have affected which medication the patients were given. However, Tables 1–3 show that patients with treatment of an unfavorable metabolic profile had a very similar proportion of the metabolic syndrome compared with the total sample.

### Comparison with the existing literature

Although several studies address the possible associations between different HTR2C gene polymorphisms and the presence of the metabolic syndrome, only the Dutch studies focus on the rs1414334 polymorphism (*Mulder et al.*, *2007*; *Mulder et al.*, *2009*; *Risselada et al.*, *2012*; *Ma et al.*, *2014*). The main statistically significant finding of these studies was a direct association between the presence of the C allele of the polymorphism and the metabolic syndrome. As our results differ from those of these other authors, we cannot consider the

association to be confirmed. Furthermore, the Dutch group did not stratify their results by gender, and since the HTR2C gene is X-linked, gender-specific analyses should have been carried out.

In agreement with *Kuzman et al.* (*2008*), it is possible that the variability of factors analyzed may influence the differences found. The prevalence of the C allele of the polymorphism analyzed in our study was similar to that of the Dutch studies. Conversely, the prevalence of the metabolic syndrome in our sample was slightly higher (*Mulder et al.*, *2007*; *Mulder et al.*, *2009*; *Risselada et al.*, *2012*; *Ma et al.*, *2014*). Additionally, previous studies did not assess atypical antipsychotics exclusively, as a percentage of the subjects had been prescribed traditional antipsychotics. Finally, the prevalence of schizophrenia in these papers was higher than in ours (*Mulder et al.*, *2007*; *Mulder et al.*, *2009*; *Risselada et al.*, *2012*). It is possible there were other differences on variables that were not reported in the Dutch studies. For example, our patients had their disease for an average of almost 15 years and had been taking the current antipsychotic medication for approximately five years. This information has not been analyzed by the Dutch group. On the other hand, there is a strong possibility that previous findings were false positives (*Ioannidis et al.*, *2001*). Therefore, the differences might be due to these factors.

### Implications for research

The set of studies by the Dutch group indicates that the presence of the C allele of the HTR2C rs1414334 gene polymorphism appears to be associated with the metabolic syndrome. Our findings, however, do not support this association. As mentioned above, this could be due to differences between our patients and those of the Dutch group or could indicate type 1 error in the original findings. To distinguish these possibilities, we would need multicentre studies evaluating the association in different antipsychotic treatments, preferably using a prospective design. However, before investing in such research, it would be helpful to establish whether this SNP has a functional influence.

## ACKNOWLEDGEMENTS

The authors thank Maria Repice and Ian Johnstone for help with the English language version of the text.

### Funding

The authors received no funding for this work.

### Competing Interests

Antonio Palazón-Bru serves as an Academic Editor for PeerJ.

### Author Contributions

- José María Rico-Gomis conceived and designed the experiments, performed the experiments, analyzed the data, wrote the paper, reviewed drafts of the paper.

- Antonio Palazón-Bru conceived and designed the experiments, analyzed the data, wrote the paper, prepared figures and/or tables, reviewed drafts of the paper.
- Irene Triano-García, Ana García-Monsalve and Andrés Navarro-Ruiz conceived and designed the experiments, performed the experiments, contributed reagents/materials/analysis tools, reviewed drafts of the paper.
- Luis Fabián Mahecha-García, Berta Villagordo-Peñalver, Jessica Jiménez-Abril and Alicia Martínez-Hortelano conceived and designed the experiments, performed the experiments, reviewed drafts of the paper.
- Vicente Francisco Gil-Guillén conceived and designed the experiments, analyzed the data, reviewed drafts of the paper.

## Human Ethics

The following information was supplied relating to ethical approvals (i.e., approving body and any reference numbers):

To comply with the current legislation, all patients were adequately informed about the study (written form) and, if agreeing to participate, signed a consent form before inclusion.

The study was approved by the Clinical Research Ethics Committee of the General University Hospital of Elche on November 20, 2012.

## Data Availability

The raw data has been uploaded as Data S1.

## Supplemental Information

Supplemental information for this article can be found online at http://dx.doi.org/10.7717/peerj.2163#supplemental-information.

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
