# Peer review of "Association between the HTR2C rs1414334 C/G gene polymorphism and the development of the metabolic syndrome in patients treated with atypical antipsychotics"

_PeerJ, doi:10.7717/peerj.2163_

## Round 0.1 · original submission · Major Revisions

The authors provide independent data on a question of interest. However, the reviewer notes several important limitations of the study that deserve more explicit recognition, and requests several items of missing data. Please also address the power analysis as discussed by the reviewer. All these points must be satisfactorily addressed and the claims limited accordingly.

Reviewer 1 ·

Basic reporting

This is a genetic association study between the rs1414334 C/G polymorphism in
the HTR2C gene and the development of the metabolic syndrome in patients treated with
atypical antipsychotics. A cross-sectional study was conducted in 166 patients treated with a variety of antipsychotics and co-medication. Potentially confounding variables were assessed and included in the statistical model. The results showed
that the presence of the C allele of the polymorphism had a protective effect in relation to
the metabolic syndrome, without reaching statistical significance (p=0.101).

Experimental design

The sample remains heterogeneous and the cross-sectional design bears important limitations, as other factors than the antipsychotic medication could have caused onset of metabolic syndrome.
The study sample is relatively small.
The type of power analyses is unusual and the power set at 99% appears somewhat unrealistic. It would be more appropriate to use standard power analyses that take allele frequencies into account.
Only one polymorphism was analyzed, although it is nowadays more common standard to cover a gene with multiple SNPs and to also conduct haplotype analyses.
The 5-HT2C gene was reported to be rather associated with weight gain in earlier stages of treatment which should be commented.
The functional influence of this SNP is not explained; is there any knowledge on this?
It appears that the results are somewhat overstated, as they were negative but are presented as opposite findings to the Dutch group.
Given that the 5HT2C gene is X-linked, gender specific analyses should be carried out.
There is no comment with respect to the ethnic Distribution of this sample.

Validity of the findings

Overall, it is rather not surprising that the findings tend to be negative and all the limitations should be addressed.

---

## Round 0.2 · Major Revisions

The Academic Editor has agreed to consider the appeal. Further evaluation of a revised manuscript is requested.

· Appeal

Appeal

Dear Editorial Committee

Please find attached our manuscript entitled Association between the HTR2C rs1414334 C/G gene polymorphism and the development of the metabolic syndrome in patients treated with atypical antipsychotics for consideration for publication in PeerJ. An earlier version of this work was sent to PeerJ (Reference #2015:10:6991) but it was rightly rejected by the editorial committee. We consider that the decision was fully justified as the manuscript failed to fulfil the standards for publication in PeerJ. However, the editor and the academic reviewers kindly offered a series of very relevant comments, all of which would help to make the article acceptable. After working on the article, we have managed to address all the comments given. Accordingly, we wish to have the study reappraised, considering that the editorial committee mentioned it was interesting but required important changes. We sincerely believe that, after addressing all the original comments, the manuscript now merits reconsideration.

Thanking you in advance, I remain

Yours

Dr Palazón-Bru (corresponding author of the manuscript)

· · Academic Editor

Reject

As the reviewer points out, the analysis of this X-linked gene needs to be done separately for men and women. Several other of the points made previously also were not adequately addressed.

Reviewer 1 ·

Basic reporting

No comment

Experimental design

No comment

Validity of the findings

No comment

Additional comments

The manuscript has been revised and improved in its quality.
However, this reviewer feels that analyses should be carried out for males and females separately, given that the 5-HT2C gene is X-linked. This is important because females encounter a complete different situation than males, and viceversa. Females can be heterozygotes whereas males can not, and this creates a completely different biological scenario. In addition, this would also be important with respect to previous articles published on this topic. In this context, figure 1 should also be revised - it is not clear to which allele (or genotype) 'polymorphism is referring to.
As indicated previouslym, this is a negative study conducted in a flawed experiment (since males and females were not analyzed separately) and as such, the general conclusion (and that results are opposite to the Dutch group) would appear misleading. The previous literature with respect to the 5-5TH2C and metabolic paramters is not well summarized neither, which could be improved.
This reviewer would recommend a point-to-point reply listing how each of the reviewers' comments have been addressed, including copy & pasting the revised sentences and paragraphs.

---

## Round 0.3 · Minor Revisions

I have been asked to consider your appeal regarding your paper "Association between the HTR2C rs1414334 C/G gene polymorphism and the development of the metabolic syndrome in patients treated with atypical antipsychotics". I am pleased to tell you that I do think the data merit publication in PeerJ, but only with some further revision and shortening.

My expertise is a long way from the topic of metabolic syndrome, but I am very interested in issues of scientific reproducibility, including in the context of genetic studies. Your study has the disadvantage of having null results based on a small sample, and my initial reaction was that the power would be too low to give a convincing result. However, I see that you were attempting to replicate a previous study that claimed a large effect on the basis of a similarly small sample, and that you did indicate that power was adequate for the replication. It was helpful to have your raw data available to get a sense of the numbers on which the conclusions were based. I note that reviewer 1 did initially recommend that the data from your study were publishable, but the paper was subsequently rejected after revision because it was judged you had not made necessary revisions; you have now dealt with these points in the latest version. I think that in the interests of counteracting publication bias that favours only positive results getting into the literature, PeerJ should publish this work, but you will need to make the following changes in order for the paper to be acceptable.

Abstract

This will need to be rewritten to take into account my comments (below) about the 55K binary logistic regressions (which I recommend removing).

The final comment:

"The results showed that the presence of the C allele of the polymorphism had a protective effect in relation to the metabolic syndrome, without reaching statistical significance (p=0.101). As in other polymorphisms, our findings showed disagreement with the previous findings of other authors. Therefore, more studies are needed to determine the association between this polymorphism and the metabolic syndrome."

alter to:

"We did not confirm previous claims that the C allele of the polymorphism was linked to metabolic syndrome: the association was in the opposite direction and non-significant. This conclusion held after taking gender and lifestyle variables into account."

My concern here, and in other parts of the paper, is that attempts are made to interpret what are likely to be spurious findings.

Introduction

You do not adequately explain why your focus was on one polymorphism of HTR2C. Some of the content from lines 224-229 is relevant here and could be moved to the introduction. I was still unclear whether rs1414334 was the only SNP where the association had replicated in previous studies, and if so, whether this meant that other SNPs had been tested but failed to replicate, or whether they had only been tested in one study. You need to give a better justification of why you selected this SNP. I think you could move some material from the Discussion section to the Introduction to make this case.

Around lines 61-63, you need also to provide information about the size of the association that was reported by the Dutch studies, and also mention the sample sizes that had been used. If the data are provided in that form, it would be good to have a very simple statement of the kind: “Among patients receiving atypical antipsychotics, x out of N (xx%) with the C allele had metabolic syndrome compared with x out of N (xx%) with the G allele.” (Provide this information for each study that tested for this effect). You should also say something like: “HTR2C is X-linked, and so females can be heterozygous whereas males have only one copy of the gene. Data from these previous studies did not, however, break down results by gender.”

line 65, last sentence reword to “We aimed to replicate the previous finding of an association between the presence of the C allele of rs1414334 and metabolic syndrome in patients treated with atypical antipsychotics. We extended previous work by considering gender effects on the association, and by incorporating measures of relevant lifestyle variables, such as diet.”

This makes clear how your study goes beyond the previous ones.

Statistical methods

The data seem massively overanalyzed. The convention on how many variables you can include in a regression in relation to number of observations serves to avoid spurious associations that arise from overfitting. You cannot escape the problem by running every permutation and combination of 6 variables and looking for the best – this has exactly the same issues – i.e. even with a completely random dataset there would be a very high chance of finding some spurious associations.

I recommend a much simpler approach, which would start by presenting 3 simple tables showing Ns for C/G status vs metabolic syndrome status for male, female and total sample. This immediately makes clear the lack of the predicted association. This is the most relevant part of the data and currently it gets lost amidst the numerous complex analyses.

You do not need the first paragraph of results, nor do you need Figures 1 and 2. I appreciate that you added Table 1 to address the reviewer concerns about gender, but it is now overkill, as it puts all the emphasis on gender, which is not the central focus of the paper. The reviewer really just wanted you to establish whether the association with genotype varied by gender, and you can show that it does not with a simple table of the kind I suggest above.

Given that you have a null result, I think it is worth briefly considering the lifestyle variables you have measured to see whether they could possibly have acted as confounders that might have masked a true association. You could at the same time report the extent to which these variables relate to the metabolic syndrome, which would give your study some validation in so far as you find associations similar to previous studies. I found it frustrating that I could not get much sense of this from the way the data were presented.

I think we therefore need either a simple table, or (if few variables are involved) a brief report in the text noting those lifestyle variables that did relate to metabolic syndrome. (Some of this is given in lines 177-8). There would then be various approaches you could adopt: the simplest might be to simply take just those variables that differentiate metabolic syndrome vs nonmetabolic syndrome cases as independent variables in a logistic regression. If there are more than 5 variables, then you could either do some data reduction by extracting factors (principal components) from correlated variables, or you could take the five with largest effect size. You could then add C/G polymorphism and see if it accounts for additional variance in predicting metabolic syndrome.

I think the other tables could be omitted, especially since you have made your raw data available, so readers who were interested in additional analyses could do this.

Discussion

The sentence on lines 196-197 should be deleted, as you have not found an inverse association – you have found a null result.
line 202-204: This sentence should be deleted, as your approach would not give more precise results – it would give results that had a very high proportion of type 1 error. For further explanation see Andrew Gelman’s work on ‘forking paths’ www.stat.columbia.edu/~gelman/research/unpublished/p_hacking.pdf
and
de Groot, A. D. (2014). The meaning of “significance” for different types of research [translated and annotated by Eric-Jan Wagenmakers, et al] Acta Psychologica, 148, 188-194. doi: http://dx.doi.org/10.1016/j.actpsy.2014.02.001
(I am not suggesting that you need to cite these sources, but I think the messages in them are very relevant for your work).

line 210-213: the material stated here sounds more like it belongs with Strengths rather than Limitations.

line 219-220: This is rather unclear. Do you mean that development of the metabolic syndrome may have affected which medication the patients were given?

line 222-223: remove this sentence

line 246: ‘we believe that’ – I would change to ‘it is possible that’.

lines 246 onwards: There is a strong possibility that previous findings were false positives. This paper by Ioannidis et al is worth reading (and citing) in this context Ioannidis, J. P. A., Ntzani, E. E., Trikalinos, T. A., & Contopoulos-Ioannidis, D. G. (2001). Replication validity of genetic association studies. Nature Genetics, 29(3), 306-309. doi: 10.1038/ng749

line 276-282. This is repetitive and should be deleted.

Minor typographical changes (to make clearer and more succinct)

Line 42: delete “it has been determined that”
Line 45: reword last sentence “Therefore identifying relevant genetic variants would be useful…”
Line 48: "Furthermore, the literature also shows that there is " revise to "Previous studies have reported"
Line 47: "Regarding the rs1414334 C/G polymorphism in the HTR2C gene, in our literature search we found only 3 studies, conducted in the Netherlands, of which two were replication studies " reword to "Our focus was on the rs1414334 C/G polymorphism in the HTR2C gene, because
Line 77 ‘treated for’
Line 117; it is unclear what is meant by ‘the scale of all the food’
Line 133-4: the word ‘exposure’ seems a bit odd when referring to a genetic variant. Reword to “C allele in 30% of cases and 13% of controls”
line 208: ‘established the existence of’ change to ‘assessed’

---

## Round 0.4 · Minor Revisions

I think the changes you have made have considerably improved the paper and I have only very minor suggestions for further modification. If you can do these, I will be happy to accept the manuscript.

Line 190. At this point in reporting p = .062, please add "(direction contrary to prediction)".

Line 239 onwards: I felt this concluding material would benefit from slight reorganisation to clarify two points a) differences could reflect population differences (some of which are known and some not); but b) could also indicate type 1 error in earlier studies. These points are rather muddled together, so I suggest reorganising as follows.

Line 240. after 'the differences found' insert the material from lines 248-250. Also the information from lines 244-247. [i.e. start with comparisons of samples on variables where you can compare]

Then say "It is possible there were other differences on variables that were not reported in the Dutch studies." and continue with the material from sentence starting in mid of line 240 to line 244.
I think the material re "Implications for research" could be shortened and also reorganised.
I suggest rewording as follows:
mid line 255 after 'Dutch group' add ', or could indicate type 1 error in the original findings. To distinguish these possibilities we would need multicentre studies evaluating the association in different antipsychotic treatments, preferably using a prospective design. However, before investing in such research, it would be helpful to establish whether this SNP has a functional influence.'
And I would leave it at that.
While you as authors clearly have the right to draw your own conclusions, I think there are real risks in recommending further association studies when it is very possible that initial findings were false positives and when we do not know if this SNP points to a biologically plausible mechanism. Hence the suggested rewording.

Finally : minor typo line 393 antipsychotic-induced

---

## Round 0.5 · accepted · Accept

I am pleased your work on this study has led to an acceptable result.